# The Microstructure Evolution of Mg-RE Alloy Produced by Reciprocating Upsetting Extrusion during Hot Compression

Ziwei Zhang 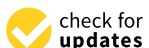, Jianmin Yu *, Zeru Wu, Hongbing Hu, Zhimin Zhang, Mo Meng, Yong Xue and Xubin Li

College of Materials Science and Engineering, North University of China, Taiyuan 030051, China; z1838674737@163.com (Z.Z.); wze9711@163.com (Z.W.); hu2920251868@163.com (H.H.); zhangzhimin@nuc.edu.cn (Z.Z.); meng19831021@163.com (M.M.); merryjukii@sohu.com (Y.X.); lixu@nuc.edu.cn (X.L.)
* Correspondence: minyu889@hotmail.com; Tel.: +86-13934224122

**Abstract:** Mg-13Gd-4Y-2Zn-0.4Zr (wt. %) alloy bar produced by three passes reciprocating upsetting extrusion (named as RUE-ed bar) exhibited fine grain with the average grain size of 3.02 μm. Hot compression tests of the RUE-ed bar were carried out on Gleeble-3800 compression unit at different deformation temperatures (653, 683, 713, and 743 K) and strain rates (0.001–1 s, 0.01–1 s, 0.1–1 s, and 0.5–1 s). This alloy showed work hardening and softening stages in hot compression, the thermal activation energy of the RUE-ed bar was 150 ± 1 kJ/mol and the constitutive equation was: $\dot{\varepsilon} = 1.80 \times 10^9 [sinh(0.0174\sigma)]^{2.47} exp\left[-\frac{150 \times 10^3}{8.314 \times T}\right]$. Numerous $Mg_5$ (Gd, Y, Zn) phase re-dissolved in α-Mg matrix appeared in the RUE-ed samples during hot compression deformation. The movement of the dislocation stimulated the re-dissolution of the $Mg_5$ (Gd, Y, Zn) phase. The re-dissolution of $Mg_5$ (Gd, Y, Zn) phase promoted texture strengthening and DRX grains growth in this experiment. In addition, the transformation and kinking of LPSO phase played an important coordinating role in the process of hot compression; 18R-LPSO was changed to 14H-LPSO phase at low strain rate while the LPSO phase kinked dominant to coordinated deformation at high strain rate.

**Keywords:** Mg-13Gd-4Y-2Zn-0.4Zr alloy; hot compression; constitutive equation; $Mg_5$ (Gd; Y; Zn) phase



## 1. Introduction

With the pursuit of lightweight materials in the automotive and aerospace fields, magnesium (Mg) alloys have received extensive attention due to their excellent density in the past two decades. As low density metallic structural materials, the average density of magnesium is approximately one third lower than that of aluminum (Al) alloys and one fifth than that of steel [1,2]. However, the special hexagonal close-packed structure of the Mg alloy greatly limits the ductility and toughness of magnesium alloy [3]. Rare earth elements can be used as an excellent magnesium alloy modifier to improve the thermal stability of magnesium alloy at high temperature, due to the formation of stable intermetallic compounds. Especially, the addition of Y and Gd introduce grain refinement strengthening, solution strengthening and dispersion strengthening into Mg alloys, and the addition of Zn into Mg-RE alloys forms Long Period Stacking Ordered (LPSO) structure. At present, the common stacking type of LPSO phase are 10H, 14H, 18R, 24R [4–6]. The strengthening of LPSO cannot be ignored in the process of regulating the mechanical properties of magnesium alloy [7]. The microstructure evolution and mechanism of Mg-Gd-Y-Zn-Zr alloys in the process of deformation have been widely investigated [8,9].

Deformation strengthening is an important method to validly improve the microstructure and mechanical properties of magnesium alloys, and plastic deformation can significantly improve the strength and toughness of magnesium alloy. Severe Plastic Deformation (SPD) technique refers to the introduction of a large strain into the process of metal deformation and obtains ultra-fine grain. This mainly includes: High-pressure Torsion (HPT) [10],



Equal-Channel Angular Pressing (ECAP) [11], Multi-Directional Forging (MDF) [12], and Reciprocating Upsetting Extrusion (RUE) [13]. In addition, the SPD technique effectively improves the plasticity, optimizes the anisotropy and lowers the strength of magnesium alloy at room temperature. Compared with other SPD methods, RUE can be widely used in industrial production to produce large billets [14,15]. Xu et al. [16,17] studied the effect of decreasing temperature RUE on microstructure and mechanical properties, and found that the average grain size of as-cast magnesium alloy could reach 6.9 μm after three passes of RUE. They also found that the Mg-8Gd-3Y-0.5Zr alloy obtained excellent mechanical properties after four passes of deformation, UTS was 320 MPa, TYS reached 258 MPa, and elongation was 11.7%. On the other hand, the microstructure evolution and mechanism of Mg alloys in the process of RUE has been widely investigated. The results indicated the strengthening mechanism in the reciprocating upsetting and extrusion process was as follows: dislocation strengthening, grain refinement, and dispersion distribution of $Mg_5$(Gd, Y, Zn) phase. The authors of [18,19] proved that the coarse massive LPSO phase twisted and gradually became broken, the inner lamellar LPSO phase dissolved, and a large number of β-$Mg_5$(Gd, Y, Zn) phases precipitated along the dynamic recrystallization grain boundaries during RUE deformation. However, the microstructure evolution law and strengthening mechanism of uniform and high-performance of Mg-RE alloys produced by RUE during subsequent deformation have not been reported. Therefore, it is of great important to figure out the hot deformation behavior of Mg-RE alloys after RUE deformation.

To date, many studies have reported on hot deformation behavior of Mg-Gd alloys. Mosadegh et al. [20] investigated the hot compressive behavior of as-extruded Mg–5Gd–2.5Nd–0.5Zn–0.5Zr alloy at a temperature range of 350–500 °C and strain rates of 0.001–1 s$^{-1}$. They reported the twining, dynamic recrystallization (DRX), and DRX-second phase interaction. Wanru Tang et al. [21] constructed a unified constitutive equation of fine-grained Mg–7Gd–5Y–1.2Nd–0.5Zr alloy based on the pyramidal <c + a> slip, tension twinning, and dynamic recrystallization (DRX). Zhimin Zhang et al. [22] found the kink of LPSO phase played an important coordination mechanism of the homogeneous Mg-13.5Gd-3.2Y-2.3Zn-0.5Zr alloy during hot compression deformation at low temperature. However, the literature on hot deformation behaviors of Mg-Gd alloys produced by RUE was limited and still need more studies to evaluate it. Therefore, it is of great concern to figure out the hot deformation behavior of Mg-Gd billets produced by RUE.

Several points are required to explore of RUE-ed samples during hot compression: (1) The stress-strain response of RUE-ed samples during deformation; (2) The evolution of second phases and the textural changes of RUE-ed samples when they are compressed under different deformation conditions. Based on the above research, this article aims to explore the microstructure and texture evolution in the subsequent deformation process of the Mg alloys produced by RUE.

In this article, the Mg-13Gd-4Y-2Zn-0.4Zr (wt. %) alloy was hot compression after repetitive upsetting and extrusion (RUE) deformation. Compared with different compression conditions, the effect of the grains size, second-phase evolution, and the texture evolution, as well as the synergy between them was discussed.

## 2. Materials and Methods

The Mg-13Gd-4Y-2Zn-0.4Zr (wt. %) alloy bar was produced by semi-continuous casting as as-cast billet. Table 1 shows the chemical composition of the as-cast billet. Firstly, the as-cast billet with d = 330 mm diameter was homogenized at 793 K for 12 h. Then, the repetitive upsetting and extrusion (RUE) was carried out on at 723 K for 3 passes to obtain uniform and fine microstructure of magnesium alloy as the initial billet. The ram speed was 2 mm·min$^{-1}$ for all RUE passes, and the samples were reheated and held at 723 K for 30 min after completing one pass (one upsetting and one extruding process were included in one pass). The reciprocating upsetting process is shown in Figure 1a. The upsetting and extrusion experiment were both carried out on a 3000 T hydraulic press at 753 K, and the extrusion ratio was 1.96. Subsequently, the compressed sample was

machined from the billet with the size of $\varnothing 10 \times 15$ mm. The location of sampling from the billet is shown in Figure 1b. The compression experiments were carried on the Gleeble-3800 compression unit (Dynamic Systems Inc, El Segundo, CA, USA). The temperatures were 653 K, 683 K, 713 K, 743 K; strain rates were 0.001 s$^{-1}$, 0.01 s$^{-1}$, 0.1 s$^{-1}$, 0.5 s$^{-1}$, respectively. Afterwards, samples were heated to the experimental temperature for 283 K·s$^{-1}$, held for 3 min, and then compressed, cooling in cold water after compression. The engineering strain was 0.8. For consistency and accuracy, two samples were performed for each temperature and strain rate.

**Table 1.** Chemical composition of Mg-Gd-Y-Zn-Zr alloy (wt. %).

| Gd | Y | Zn | Zr | Si | Cu | Fe | Mg |
|---|---|---|---|---|---|---|---|
| 12.92 | 4.20 | 2.28 | 0.36 | <0.01 | <0.01 | <0.01 | Bal |

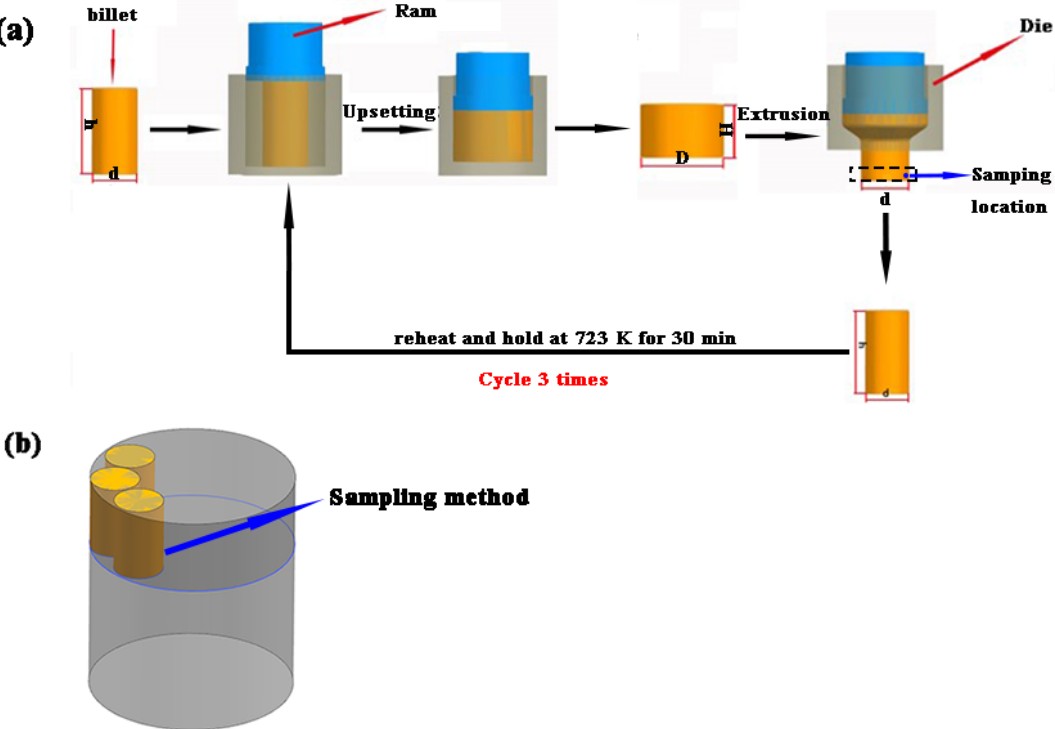

**Figure 1.** Preparation of materials: (**a**) Principle of RUE experiment; (**b**) The sampling method and position after the repetitive upsetting and extrusion (RUE).

For investigation of the microstructure of the initial samples and deformed samples, the samples were grinded by sandpapers of 1000, 3000, 5000, and 7000, and polished to obtain a smooth mirror surface. Optical microscope (OM; DM2500M, Leica Microsystems, Wetzlar, Germany) was taken to observe the microstructure after etching. The ratio of etchant was 1 g picric acid + 2 mL acetic acid + 2 mL distilled water + 14 mL ethanol, etched of the sample for about 3 s. The X-ray diffraction (XRD; Rigaku D/MAX2500PC, Rigaku, Tokyo, Japan) analysis was carried out to analyze the composition of phases in Mg-13Gd-4Y-2Zn-0.4Zr Mg alloy. Smart-Lab was used for XRD analysis at the working voltage of 50 kV, working current was 20 mA, the diffraction angle was 10–90°, the scanning step was 0.01°, and the scanning speed was 10°·min$^{-1}$. MDI Jade 6.0 software was used to compare and calibrate the materials, and the phase composition was determined. These samples were taken to further analysis of phase using a scanning electron microscope (SEM; SU5000, Hitachi, Tokyo, Japan) technique under working voltage of 50 kV and working distance of 10 mm. The electron backscatter diffraction analyzer (EBSD, EDAX Inc., Mahwah, NJ,

USA) was carried out to observe the texture. The working voltage was 70 kV and the working distance was 15 mm. The observation direction of OM, SEM, EBSD, XRD were perpendicular to the ED (extrusion direction)–TD (transversal direction) plane.

## 3. Results and Discussion

### 3.1. Initial Microstructure

The initial microstructure was shown in Figure 2. The observation direction was shown as the red area in Figure 2. It was observed that the RUE-ed samples consisted of dynamic recrystallization grains, LPSO phases, and the precipitates (The small black particles at the grain boundaries and inside the grains). The LPSO phases were divided into a lamellar phase and block-shaped phase as shown in Figure 2a. This microstructure was also observed in our previous research [17,19]. The fine DRX grains and the precipitates inside the alloy were difficult to distinguish. To clearly view the existence of the precipitates, SEM was utilized. A great deal of cell-shaped precipitates in grain interiors and on grain boundaries was evidenced with SEM-BSE images and XRD analysis (Figures 2b and 3c). On the other hand, this showed that a discontinuous dynamic precipitation behavior deformed in grain interiors and on grain boundaries during RUE deformation.

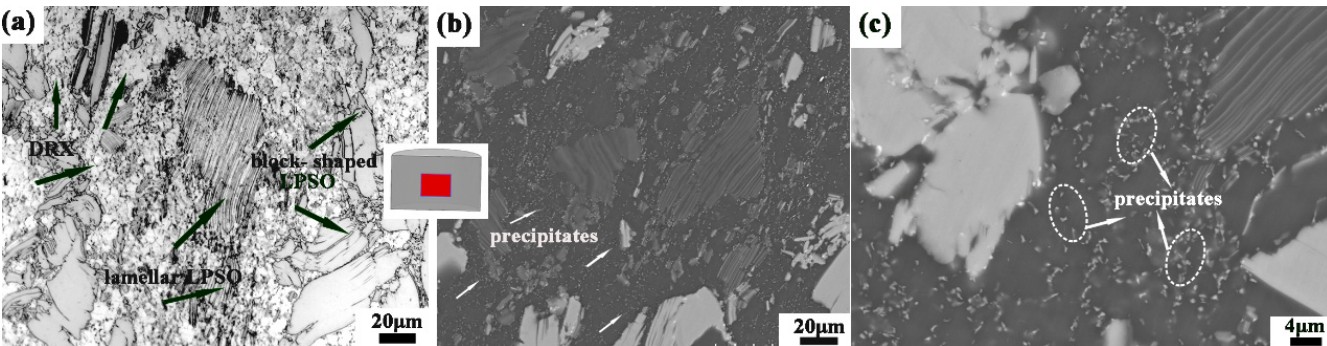

**Figure 2.** OM and SEM-BSE images of the initial billet: (**a**) OM map; (**b**,**c**) SEM-BSE maps.

EBSD analysis paralleled to the loading direction of the initial billet is presented in Figure 3. The change of grain orientation can be observed by the change of color in Figure 3a. Thus, it can be noticed that small orientation differences were inside grains due to the existence of sub-grain boundaries. Moreover, it was observed that the grains were very fine, and the average grain size was 3.02 μm as shown in Figure 3b. In addition, XRD analysis showed α-Mg, $Mg_5$ (Gd, Y, Zn) and LPSO phases (Figure 3c) [16–18]. It can be noted that weaker texture was deformed inside of the RUE-ed samples but not the traditional extrusion texture and {0001} pole figure suggested that the maximum basal intensity was 1.73 (Figure 3d). In previous investigations, the microstructure of RUE Mg alloys was reported as DRX grains and deformed grains. Similarly, pole figure (PF) analysis in Figure 4 showed the basal texture of the deformed grains was stronger than the DRX grains and the maximum intensity of (0001) basal texture was 4.71. The basal planes were paralleled to the direction of extrusion (ED) but the maximum intensity of (0001) basal texture was in the range of 0–30° near the equator.

In addition, PF analysis in Figure 4f shows the basal texture of DRX grains with a texture similar to bimodal type, and the basal texture strength was just 1.99. Additionally, it also shows the basal planes of the most DRX grains were perpendicular to the ED direction, but the maximum inclination angle was ~40° with respect to ED direction. Therefore, it can be suggested that the texture of the alloy was weakened due to the occurrence of dynamic recrystallization during RUE deformation, therefore, the plastic deformation capacity of the material was improved.

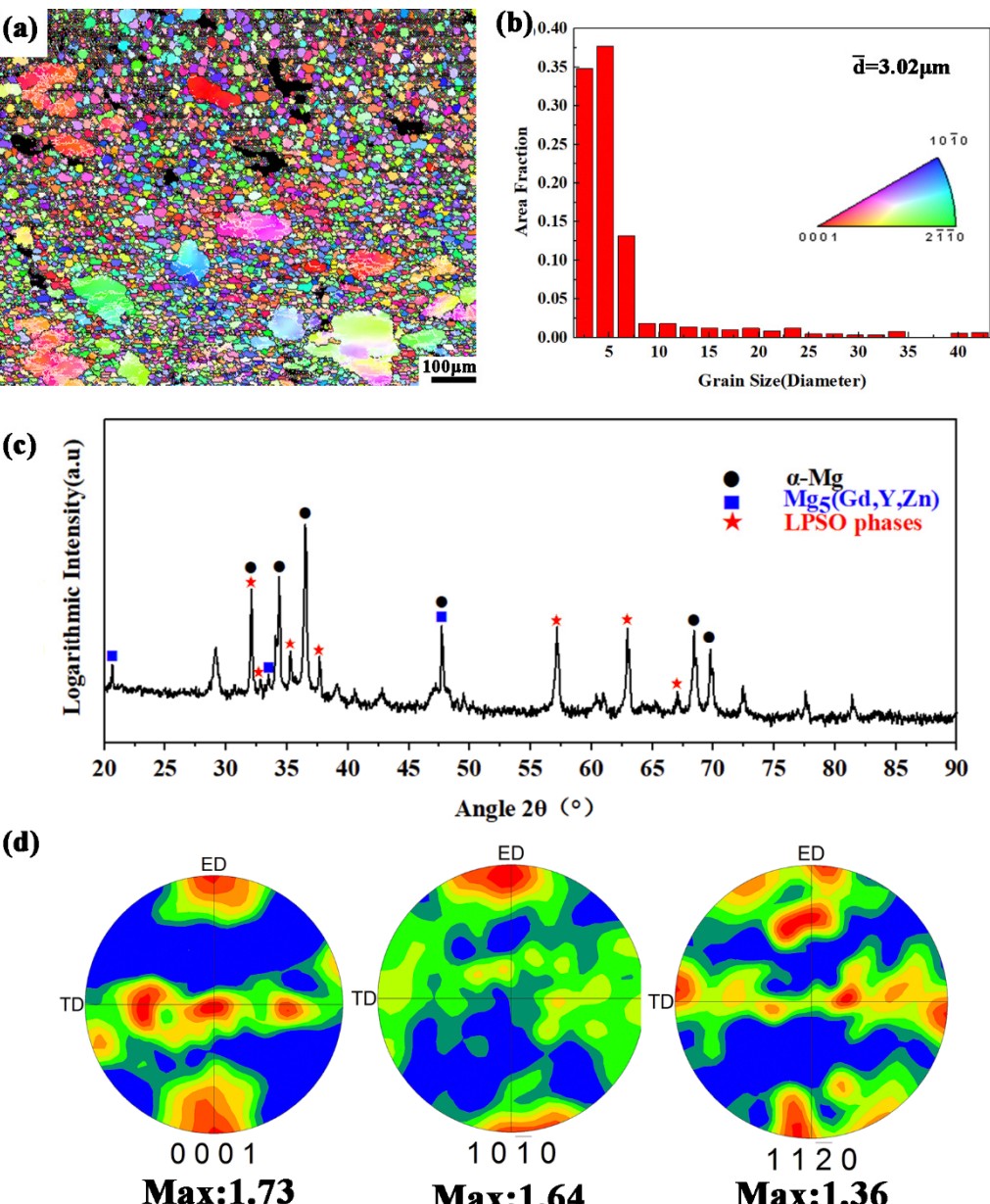

**Figure 3.** Microstructure of initial Mg-13Gd-4Y-2Zn-0.4Zr billet: (**a**) EBSD map; (**b**) grain size statistical analysis; (**c**) XRD analysis; (**d**) Pole figure analysis.

*3.2. Results of Compression Experimental*

3.2.1. Stress-Strain Response and the Establishment of Constitutive Model

　　Figure 5 was the true stress-true strain curve after thermal simulation compression under different deformation conditions. According to the stress-strain curve (Figure 5), the samples showed two obvious stages during the compression process: work hardening stage and softening stage. Evidently, true stress increased rapidly with the increase of true strain until it reached the peak stress, and softening came to dominant, which meant the true stress curve decreased with the increase of true strain. Then, the rheological stress curve gradually tended to be stable, the work hardening and softening reached a dynamic equilibrium state. Beyond that, the lower the temperature, the higher the peak stress when the strain rate ($\dot{\varepsilon}$) was the same. Additionally, at the same temperature, the peak stress reduced down with the decrease of strain rate.

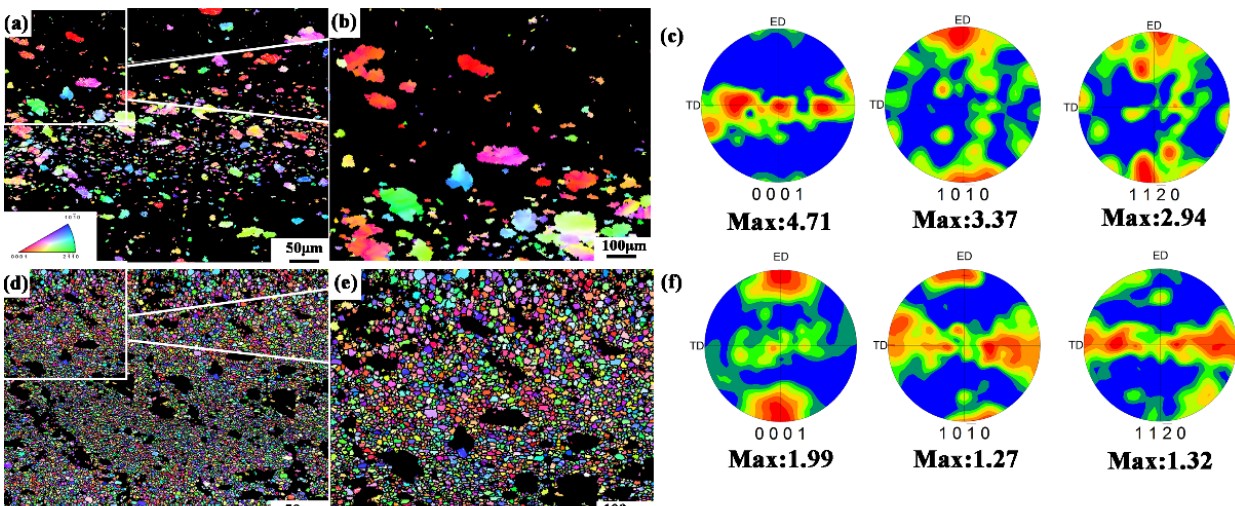

**Figure 4.** EBSD analysis of initial Mg-13Gd-4Y-2Zn-0.4Zr billet: (**a**,**b**) EBSD map of deformed grains (**d**,**e**); EBSD map of DRX grains; (**c**) Pole figure (PF) analysis of deformed grains; (**f**) Pole figure (PF) analysis of DRX grains. (**b**,**e**) are partial enlarged views of the white square frame areas in (**a**,**d**), respectively.

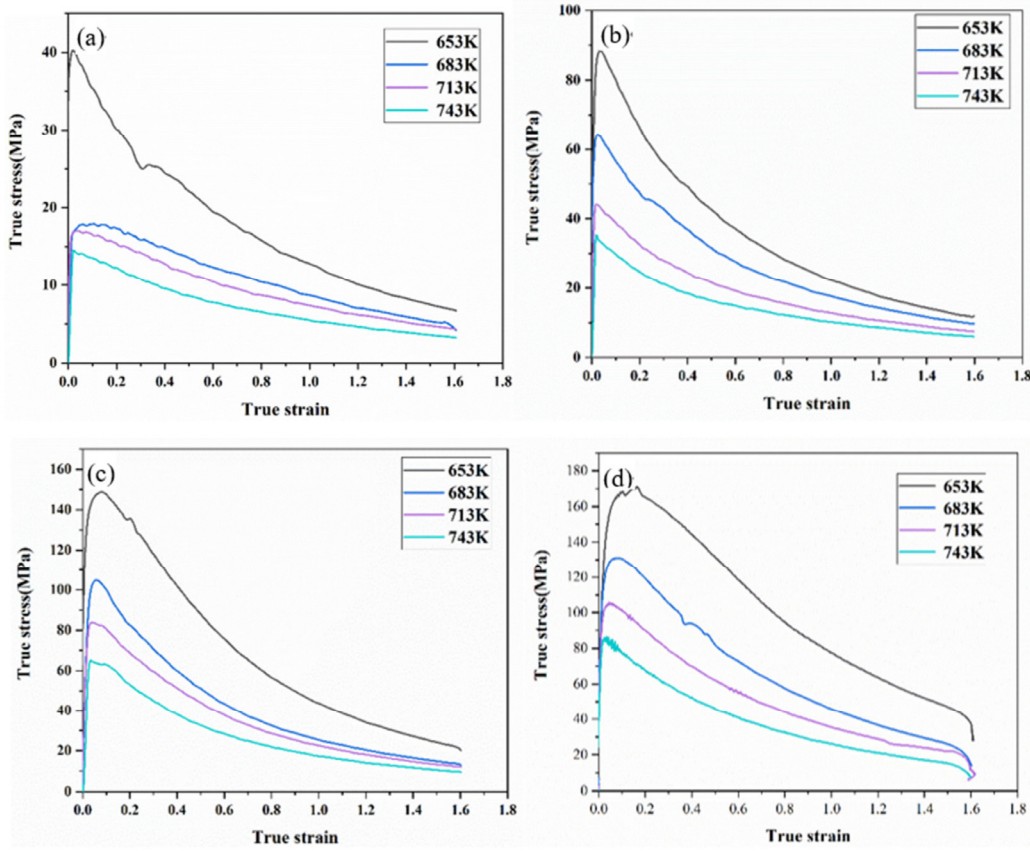

**Figure 5.** Stress-strain curves under different compression conditions of Mg alloy: (**a**) $\dot{\varepsilon}$ = 0.001 s$^{-1}$; (**b**) $\dot{\varepsilon}$ = 0.01 s$^{-1}$; (**c**) $\dot{\varepsilon}$ = 0.1 s$^{-1}$; (**d**) $\dot{\varepsilon}$ = 0.5 s$^{-1}$.

The thermal deformation dynamic model was instructive to production practice, so the relationship among peak stress ($\sigma_p$), deformation temperature, and strain rate were analyzed to establish constitutive model. The peak stresses under different deformation conditions are shown in Table 2.

**Table 2.** The peak stress ($\sigma_p$) under different deformation conditions.

| Strain Rate (s $^{-1}$) | Temperature (K) | | | |
|---|---|---|---|---|
| | 653 | 683 | 713 | 743 |
| 0.001 | 40.2 ± 0.03 | 17.94 ± 0.01 | 17.10 ± 0.02 | 14.54 ± 0.02 |
| 0.01 | 88.62 ± 0.02 | 64.33 ± 0.03 | 44.24 ± 0.06 | 35.21 ± 0.05 |
| 0.1 | 148.85 ± 0.01 | 104.84 ± 0.1 | 83.7 ± 0.1 | 64.56 ± 0.02 |
| 0.5 | 171 ± 0.45 | 130.03 ± 0.01 | 105.6 ± 0.5 | 85.6 ± 0.2 |

The constitutive equation was established based on the modified hyperbolic sine function Arrhenius [21] relation.

$$\dot{\varepsilon}_1 = A_1\sigma^{n_1}exp[-Q/(RT)] \tag{1}$$

$$\dot{\varepsilon}_2 = A_2[exp(\beta\sigma)]exp[-Q/RT] \tag{2}$$

$$\dot{\varepsilon} = A[sinh(\alpha\sigma)]^n exp[-Q/(RT)] \tag{3}$$

Equation (1) was suitable for low stress state, Equation (2) was suitable for high stress state, and Equation (3) represented the modified hyperbolic sine function Arrhenius [21] relation to describe the relationship among the $\sigma$, $\varepsilon$, and $T$ in the process of deformation under arbitrary stress conditions, and unified in the form of Equations (1) and (2). In the above Equation, $\dot{\varepsilon}$ was the strain rate; $\sigma$ was the flow stress; $A$, $A_1$, $A_2$, $\alpha$, and n were all material constants, where $A_1 = A/2^{n_1}$, $A_2 = A\alpha^{n_1}$, $\alpha = \beta/n_1$; $Q$ was the activation energy of deformation, which was an important index to measure the difficulty of alloy deformation and indicated the energy of the customer service required for atom transition. $T$ was the absolute temperature; $R$ = 8.314 J/(mol·K) was the molar gas constant.

The Equations (1)–(3) were taken the natural logarithm respectively, and obtained Equations (4)–(6).

$$ln\dot{\varepsilon}_1\dot{\varepsilon} = n_1 ln\sigma + lnA_1 - Q/RT \tag{4}$$

$$ln\dot{\varepsilon}_2 = \beta\sigma + lnA_2 - Q/RT \tag{5}$$

$$ln\dot{\varepsilon} = nln[sinh(\beta\sigma)] + lnA - Q/RT \tag{6}$$

Therefore, it can be found that $ln\dot{\varepsilon} - ln\sigma$ were linearly correlated, and the $ln\dot{\varepsilon} - ln\sigma$ and $ln\dot{\varepsilon} - ln[sinh(\beta\sigma)]$ were both linearly correlated. According to Figure 6a, $\beta$ = 0.0593. The linear fitting results of the $ln\dot{\varepsilon} - ln\sigma$ are shown in Figure 5b and were clearly $n_1$ = 3.41 through calculation. Through this means, $\alpha = \beta/n_1$ = 0.0174 was obtained.

Assuming that Q did not change with the change of temperature, by transforming Equation (6), the calculation equation of deformation activation energy was shown as Equation (7).

$$Q = R\left\{ \frac{\partial ln\dot{\varepsilon}}{\partial ln[sinh(\beta\sigma)]} \right\}_T \left\{ \frac{\partial ln[sinh(\beta\sigma)]}{\partial(100/T)} \right\}_{\dot{\varepsilon}} \tag{7}$$

Placing $\alpha$ into Equation (7), the linear approximation of $ln\dot{\varepsilon} - ln[sinh(\beta\sigma)]$ and $ln[sinh(\alpha\sigma)] - 1000/T$ were shown in Figure 6c,d. It can be calculated that the average linear slope of $ln[sinh(\alpha\sigma)] - 1000/T$ was k = 7.17 when the strain rate was constant. The average linear slope of $ln\dot{\varepsilon} - ln[sinh(\beta\sigma)]$ was $n_{Arr}$ = 2.53 when the strain rate was constant ($n_{Arr}$ was the material constants calculated on the base of modified hyperbolic sine function Arrhenius relation), and the assumption that $Q$ did not change with the change of temperature). Then, according to Equation (7), the thermal activation energy Q = 150 ± 1 kJ/mol under different temperatures and different strain rates. Xiangsheng Xia et al. [23] calculated that the thermal activation energy of Mg–8.1Gd–4.5Y–0.3Zr alloy was 192.568 kJ/mol. Jiabin Fan et al. [24] investigated Q = 287.38 kJ/mol. Meng Mu et al. [25] studied the thermal activation energy of Mg-13Gd-4Y-2Zn-0.5Zr on stretching which was 259.13 kJ/mol, and the result (150 ± 1 kJ/mol) in this experiment was lower than previous research.

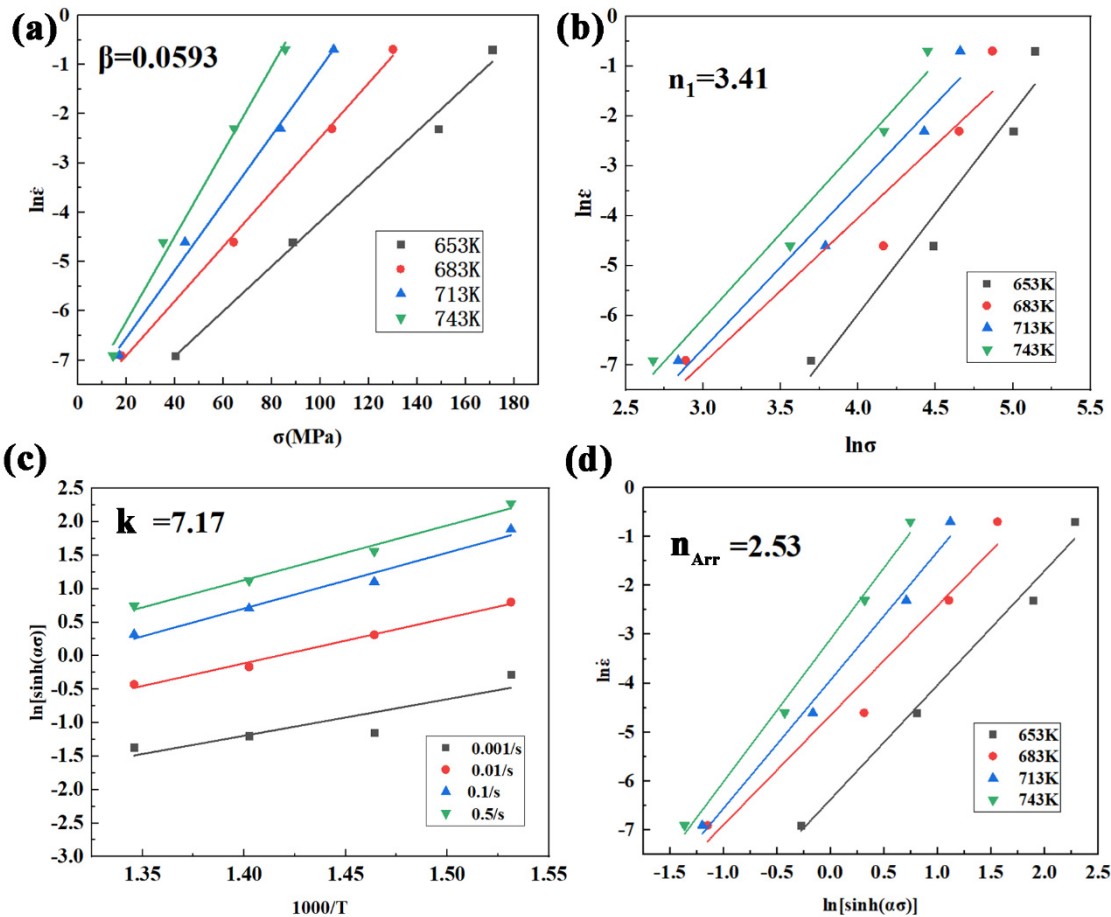

**Figure 6.** The linear fitting curves: (**a**) $ln\dot{\varepsilon} - ln\sigma$; (**b**) $ln\dot{\varepsilon} - ln\sigma$; (**c**) $ln\dot{\varepsilon} - ln[sinh(\beta\sigma)]$; (**d**) $ln[sinh(\alpha\sigma)] - 1000/T$.

Considering the effect of compression temperature and strain rate on the microstructure and properties of the alloy, the Zener–Hollomon parameter (Z) of temperature compensation was introduced. The relationship between the parameter Z and the temperature and rate of deformation was as follows:

$$Z = \dot{\varepsilon}exp(Q/RT) = A[sinh(\alpha\sigma)]^{n} \tag{8}$$

Parameter Z was obtained by taking the natural logarithm of Equation (8) as Equation (9).

$$lnZ = lnA + nln[sinh(\alpha\sigma)] \tag{9}$$

Naturally, $lnZ - ln[sinh(\alpha\sigma)]$ was linearly correlated and the fitting curve was shown in Figure 7, so it can be calculated that $n_{ZH}$ = 2.47 A = 1.80 × 109 ($n_{ZH}$ was the material constants calculated by considering the effect of the temperature and strain using the Zener–Hollomon parameter). Then, by substituting the above values (n, $\alpha$, A, Z, Q) into Equation (3), the constitutive equation was:

$$\dot{\varepsilon} = 1.80 \times 10^{9}[sinh(0.0174\sigma)]^{2.47}exp\left[-\frac{150 \times 10^{3}}{8.314 \times T}\right] \tag{10}$$

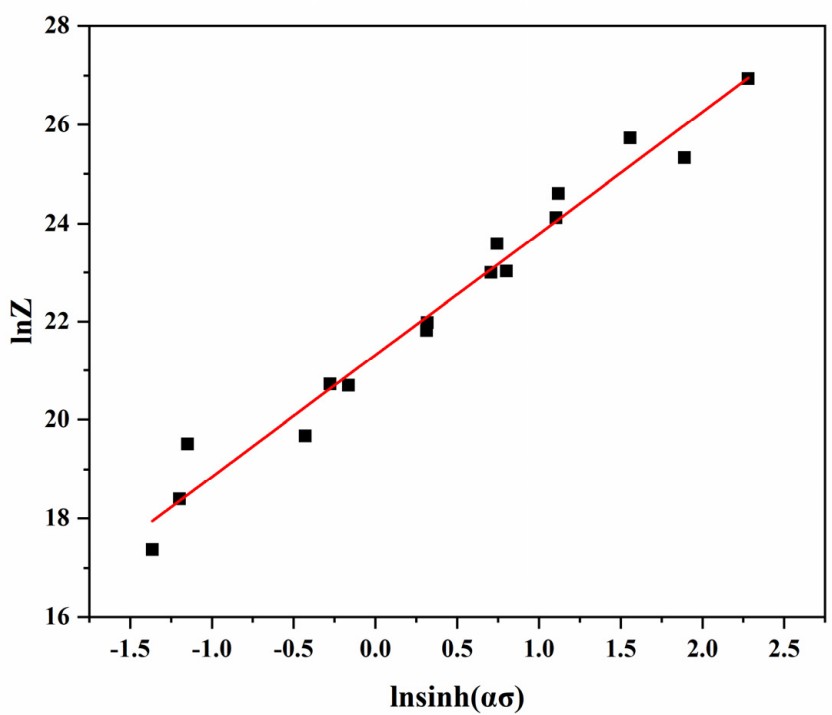

**Figure 7.** The linear fitting curve of $lnZ - ln[sinh(\alpha\sigma)]$.

3.2.2. Second-Phase Re-Dissolution of the Alloy under Different Deformation Conditions

The OM and SEM maps of the deformed specimens under different deformation conditions are shown in Figure 8. The granular precipitation decreased, and the dynamic recrystallization grain grew up with the increasing of temperature. At low temperature (653 K), a large number of precipitation particles were mainly distributed at grain boundaries and within the DRX grains. This distribution was similar to the initial microstructure in Figure 2a,b. Along with the increase of the deformation temperature, the granular precipitation distributing on the fine DRX grains boundaries decreased firstly as shown in Figure 8b,f. However, the DRX grains growth was not obvious. At high temperature (713 K), it was observed that almost all the granular precipitation disappeared with the DRX grains growing up (Figure 8c,g). As was already discussed, precipitation at grain boundaries played a great role in hindering the growth of grains. Therefore, it also supported that the growing up of the DRX grains was closely related to the re-dissolution of the precipitated phase. Thus, the precipitates experienced re-dissolution with the increase of the deformation temperature, evidenced in Figure 8a–d.

Figure 9 shows OM and SEM maps of deformed samples at a strain rate 0.5 s$^{-1}$ and at temperatures 653 K, and 743 K. The comparison of the strain rate-depended behaviors of the granular precipitation, re-dissolution, and the grain growth showed that the precipitated phase re-dissolved faster and the DRX grains which grew up faster at high strain rate. In addition, it was difficult for the granular precipitation to re-dissolve within the breaking LPSO phases. The reasons will be discussed in the following sections.

The phenomenon of the precipitated phase re-dissolution is shown in Figure 10, which was the XRD analysis of deformed samples under different deformation conditions. The width and peak of the peak shape (Mg$_5$ (Gd, Y, Zn) phase) narrowed under different temperature (0.001 s$^{-1}$). This result was corresponding to the above-mentioned phenomenon of precipitated phase re-dissolution. Meanwhile, it can be concluded that Mg$_5$ (Gd, Y, Zn) phase re-dissolved in the process of thermal compression.

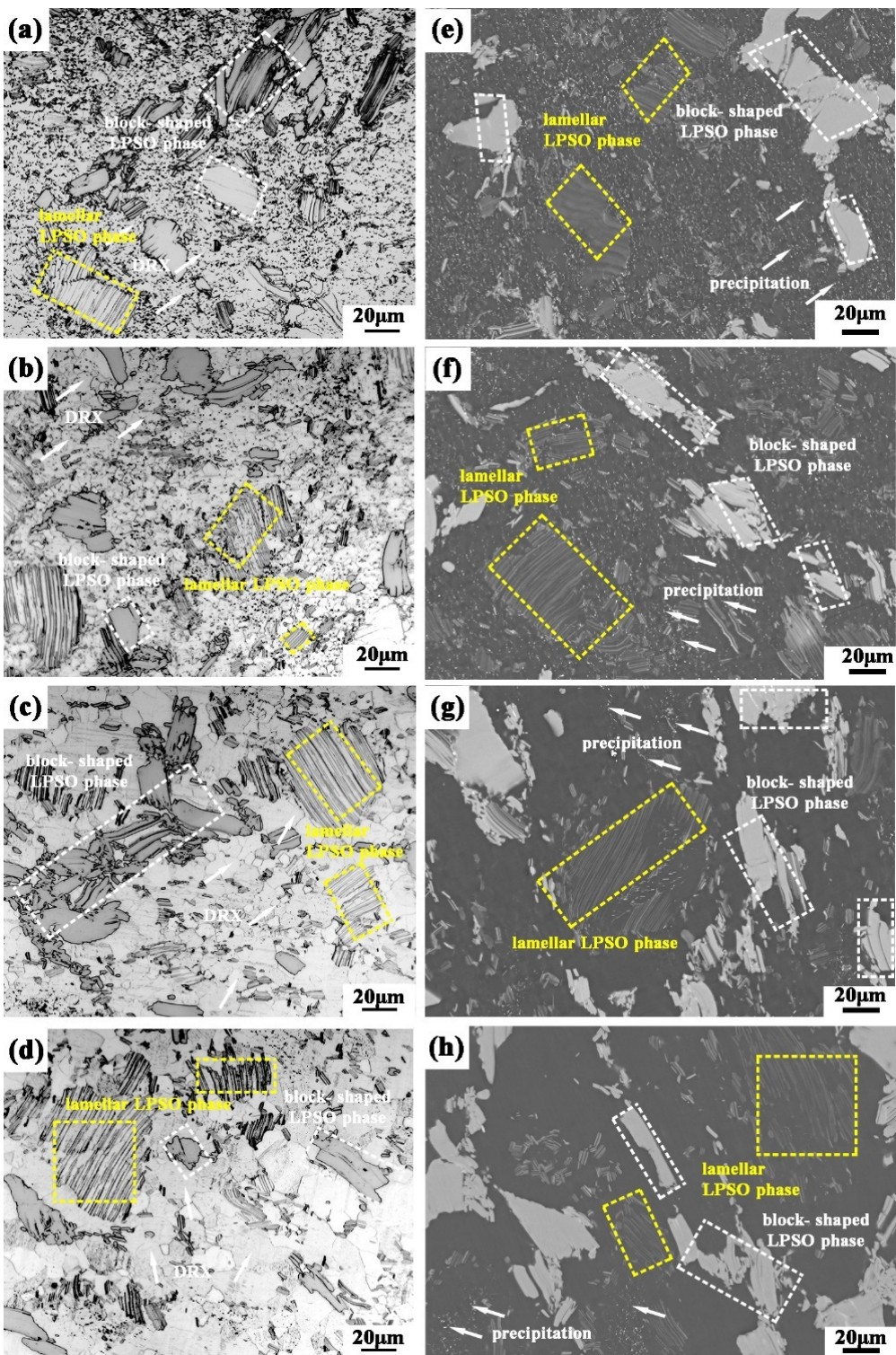

**Figure 8.** Microstructure evolution of deformed samples at a strain rate 0.001 s$^{-1}$ and at temperatures 653 K, 683 K, 713 K, and, 743 K, respectively: (**a**–**d**) OM maps; (**e**–**h**) SEM maps (white and yellow square frame showed lamellar LPSO phase and block-shaped LPSO phase, respectively).

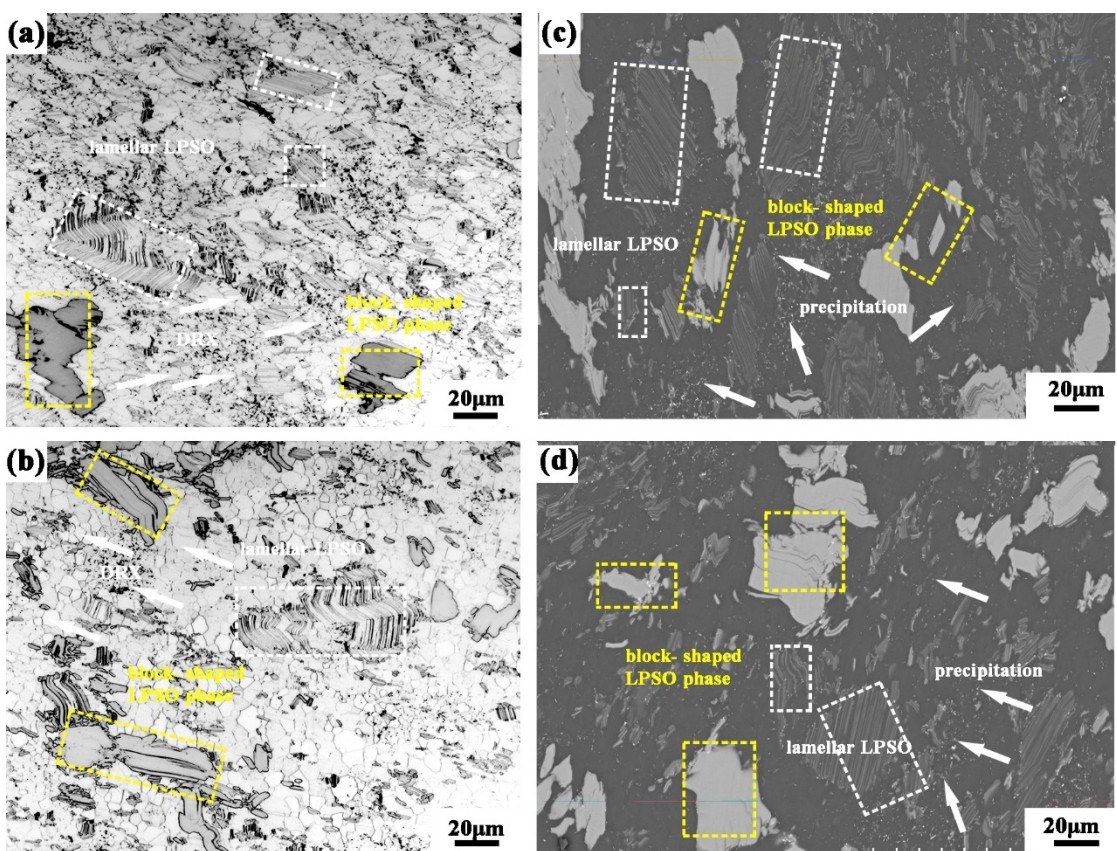

**Figure 9.** Microstructure evolution of deformed samples at a strain rate 0.5 s$^{-1}$ and at temperatures 653 K, and 743 K, respectively: (**a**,**b**) OM; (**c**,**d**) SEM maps (white and yellow square frame showed lamellar LPSO phase and block-shaped LPSO phase, respectively).

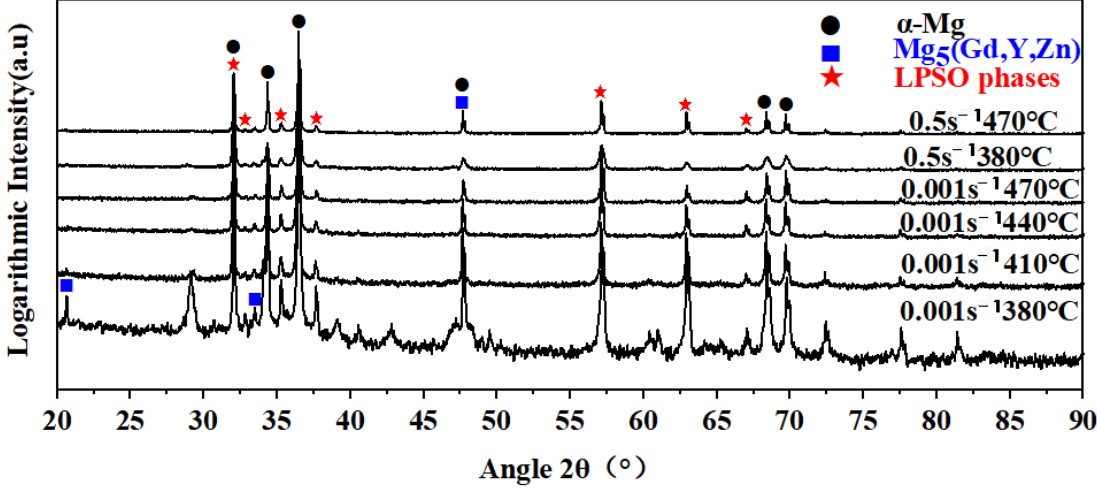

**Figure 10.** XRD patterns of deformed samples under different deformation conditions.

### 3.2.3. Evolution of LPSO Phase

K. Hagihara et al. [26] had already discussed and reported that the LPSO phases played a role to coordinate the large plastic strain by kinking, which will improve the plasticity of the alloy and further increases the deformation amount to refine the Mg matrix, thus improving the strength of the alloy. Therefore, the problem was how the evolution of LPSO phases during compression deformation. It was not arbitrary to kink for the rigid block-shaped LPSO phase (18R-LPSO) [27,28], but 18R-LPSO was changed to lamellar

LPSO phase (14H-LPSO) [29,30], and the break of the LPSO phase is shown in the blue square frame in Figure 11a. This result was basically the same as the statement of part of 18R-LPSO phase change to 14H-LPSO [31–33]. On the contrary, many interesting kink bands were observed at temperature 653 K at big strain rate (0.5 s$^{-1}$), that suggested the deformation of magnesium alloy mainly depended on the twist of layered LPSO phase to coordinate the deformation at high strain rate. K. Hagihara [34] et al. suggested the LPSO phase only had the (0001) <11$\bar{2}$0> base slip system, and the stress concentration was easy to generate at the interface between LPSO phase and α-Mg matrix due to the elastic modulus of LPSO structure which was higher than that of α-Mg matrix. At high strain rate, numerous dislocations generated at the interface between 14H-LPSO phase and α-Mg matrix led to the accumulation of dislocation, and formed the area of strong stress concentration promoting the generation of kink [35,36]. In the experiment, the stress concentration cannot be eliminated completely by dynamic recrystallization, but 14H-LPSO phase consumed part of the dislocation energy by forming a high-density dislocation kink band, so the kink deformation was the main deformation formula of the LPSO phase to coordinated deformation at high strain rate as the yellow square frame, which is displayed in Figure 11b.

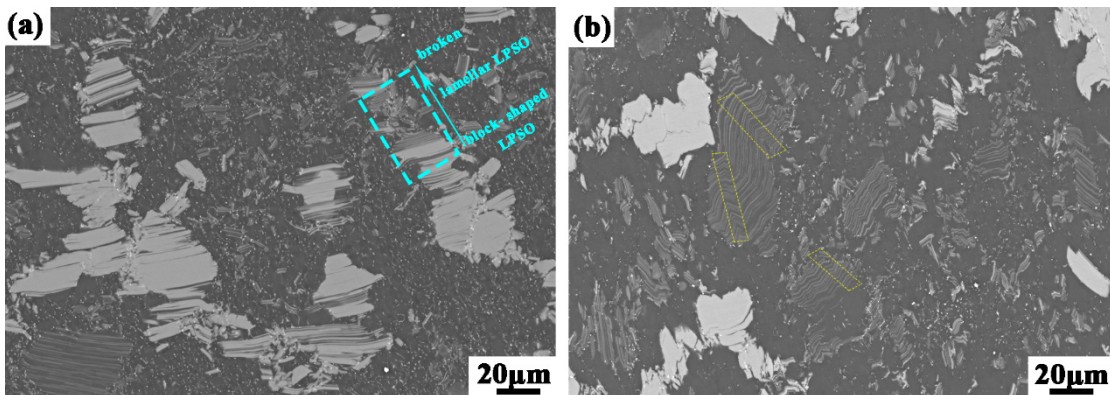

**Figure 11.** SEM maps of deformed samples at temperatures 653 K: (**a**) strain rate 0.001 s$^{-1}$; (**b**) strain rate 0.5 s$^{-1}$ (yellow square frame showed kinking, blue square frame showed the evolution of LPSO).

### 3.2.4. Texture Analysis

Samples with different deformed states characterized by EBSD: sample deformed at strain rate 0.001 s$^{-1}$ and 653 K (named as L-653), at strain rate 0.001 s$^{-1}$ and 743 K (named as L-743), at strain rate 0.5 s$^{-1}$ and 743 K (named as H-743) (L means low strain rate and H means high strain rate). Figure 12a–c shows the inverse pole figure (IPF) maps of the samples deformed at strain rate 0.001 s$^{-1}$, 0.5 s$^{-1}$, and 653 K and 743 K parallel to ED, respectively. The black area was the unrecognized phase. The corresponding average grain size of the DRX grains was displayed in Figure 11a–c. As the temperature increased to 743 K, the average grain size increased to 7.22 μm while the area fraction of DRX grains kept almost constant, 0.756 and 0.790, respectively (Figure 12a,b). The average grain size was increased with increasing of the strain rate as displayed in Figure 12a,c.

The plastic deformation of RUE-ed samples caused the development of crystallographic texture and changed the diffraction intensity of the basal planes. Initial texture exhibited that basal planes of the most DRX grains were perpendicular to the ED direction and the highest maximal intensity was with a tilt of ~40° as shown in Figure 4c. Figure 12d–f showed the samples that were subjected to uniaxial compression under different conditions. L-653 sample formed a weaker texture due to the orientation of DRX grains which was randomly distributed, with a maximum intensity 2.07 (Figure 12d). (Due to the number of grains, the statistics were consistent and the regional characteristics were basically the same, therefore, the regional error can be neglected). While at 653 K (Figure 12e), a typical {10$\bar{1}$0} <0001> texture, where basal plane was parallel to ED direction and the c-axis were perfectly

perpendicular to ED direction, was developed with a maximum intensity 5.48. On the other hand, H-743 sample showed a weaker basal texture compared with L-743 sample. Therefore, the deformed grains and the DRX grains were studied separately to expound the above phenomenon. The deformed grains possessed stronger basal texture, while orientation distribution of the DRX grains became more concentrated. Compared with Figure 4, the intensity of (0001) basal texture improved but the texture orientation was more random as both the deformed grains and DRX grains exclude the texture of the DRX grains of L-743 sample in Figure 13.

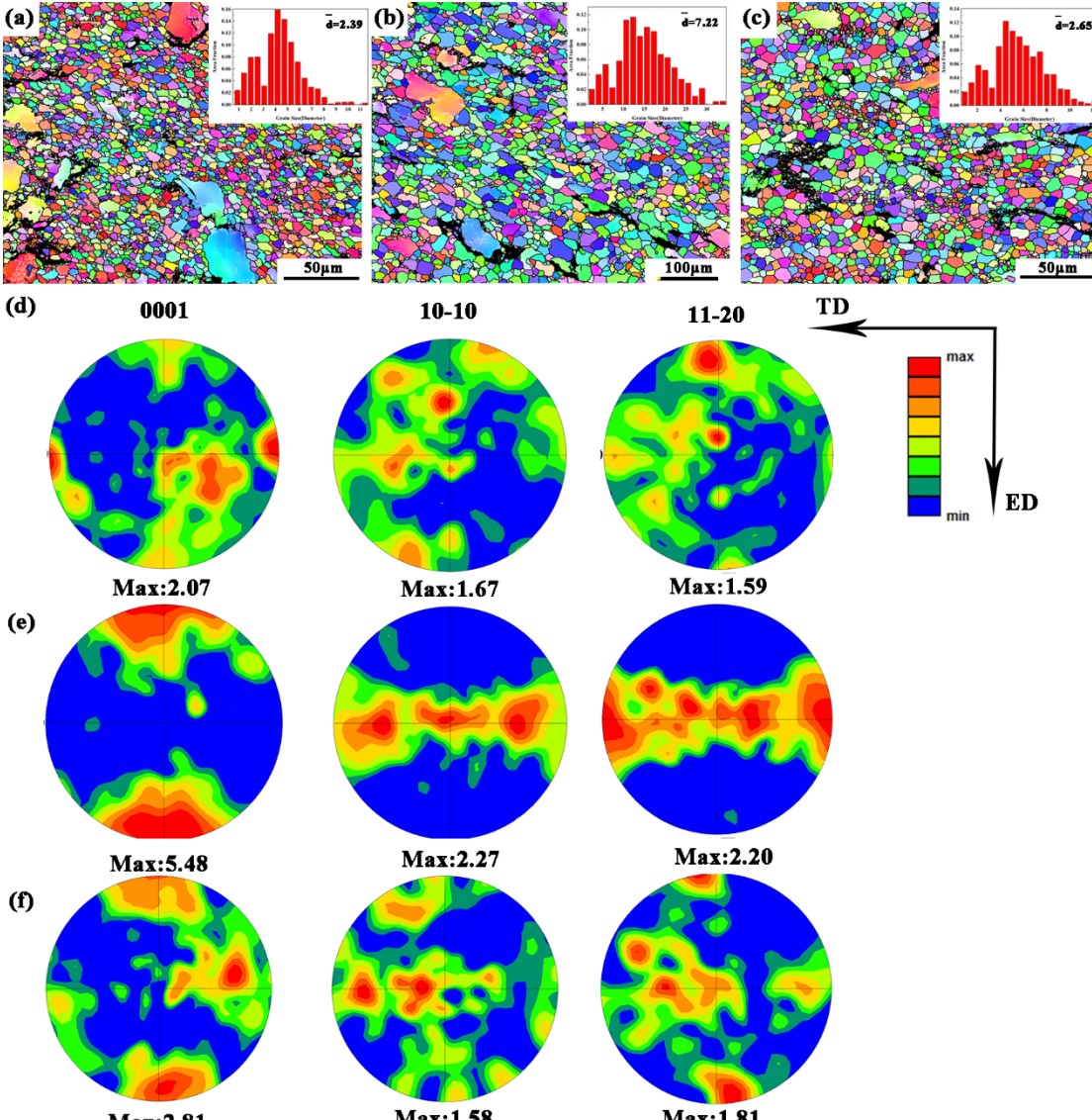

**Figure 12.** EBSD analysis and average grain size of DRX grains of the compressed Mg-13Gd-4Y-2Zn-0.4Zr (wt. %) samples parallel to ED direction: (**a**) L-653 sample; (**b**) L-743 sample; (**c**) H-743 sample (HAGBs, 15°< misorientation < 100°) and low angle grain boundaries (LAGBs, 2° < misorientation < 15°) are marked by black and white lines, respectively. Pole figure (PF) analysis of DRX grains of the compressed Mg-13Gd-4Y-2Zn-0.4Zr (wt. %) samples: (**d**) L-653 sample; (**e**) L-743 sample; (**f**) H-743 sample.

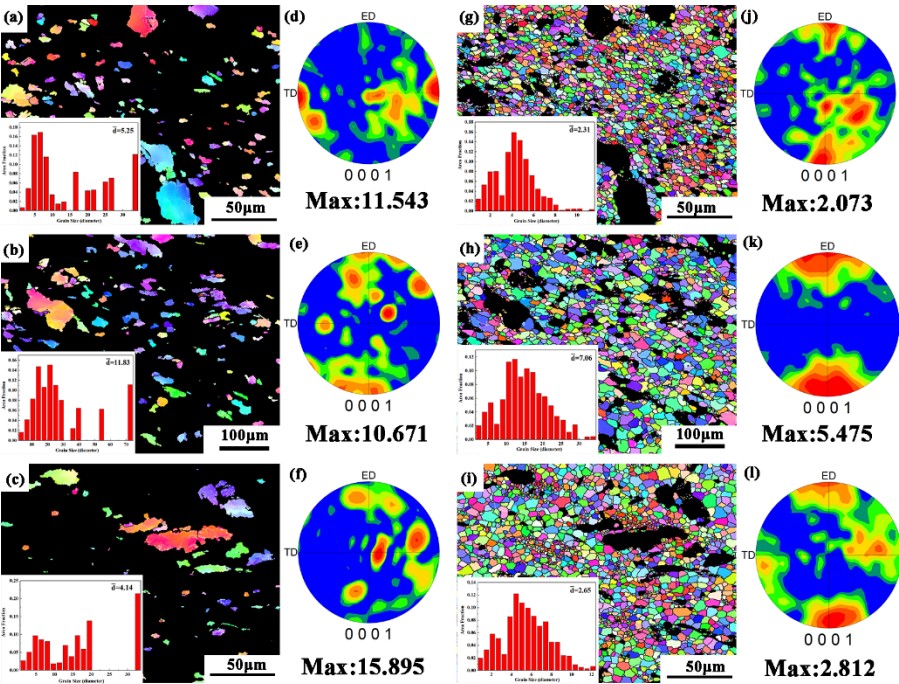

**Figure 13.** EBSD analysis and Pole figure (PF) analysis of the deformed grains: (**a**,**d**) L-653 sample; (**b**,**e**) L-743 sample; (**c**,**f**) H-743 sample. EBSD analysis and Pole figure (PF) analysis of the DRX grains: (**g**,**j**) L-653 sample; (**h**,**k**) L-743 sample; (**i**,**l**) H-743 sample.

Figure 14 shows the mean misorientation angle (MA) value of different compressed samples, which reflected the inclination degree of individual grains relative to main orientation. Figure 14 also indicates the average misorientation angle was decreased while the deformation temperature was increased at low strain rate ($0.001 \text{ s}^{-1}$). When the samples deformed at $0.001 \text{ s}^{-1}$, the average misorientation angle of hot compression deformation at high temperature was 3° lower than at low temperature (Figure 14b,c). In the same way, when the samples deformed at high temperature (743 K); the average misorientation angle of deformation samples at high strain rate was 5° higher than that at low strain rate Figure 14c,d. Moreover, it was well known that the dispersed second phase can pin the grain boundary, so that inhibited the movement of the grain boundary. However, the precipitation re-dissolved with the increasing of temperature (Figure 8). The result was against the high-energy grain boundary theory (HE) [37], which had pointed out that when the misorientation angle was 20–45°, the diffusion rate of grain boundaries will increase, and then the ability of precipitates to hinder grain boundary movement would decrease.

The kernel average misorientation (KAM) value of different compressed samples are displayed in Figure 15, showing hot compressed reduced the mean KAM value of this samples, which indicated the reduction of the dislocations and the release of local stress concentrations under hot compression. Obviously, the re-dissolution of $Mg_5$ (Gd, Y, Zn) phase weakened the ability to hinder dislocation movement, thus reducing the local stress concentration in both deformed grains and DRX grains at the same deformation strain rate. However, an interesting phenomenon was found: the mean KAM of H-743 sample was higher than L-743 sample. This showed the ability of 14H-LPSO phase kinking to consume dislocations was limited, and the rate of dislocation generation at high strain rate was greater than that consumed at twist. In summary, the second phase dissolution was the main reason for the weakening of stress concentration at low strain rate ($0.001 \text{ s}^{-1}$), while the increase of dislocation caused by the increase of strain rate was dominant at high strain rate. The decrease of local stress concentration due to the $Mg_5$ (Gd, Y, Zn) phase re-dissolution and the increase of dislocation at high strain rate should be considered simultaneously in the study of hot compression deformation.

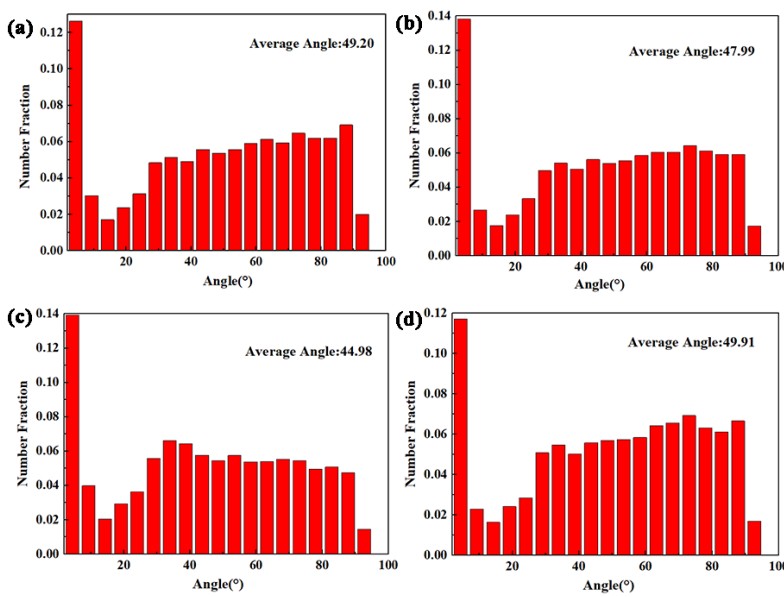

**Figure 14.** Mean MA value of the compressed Mg-13Gd-4Y-2Zn-0.4Zr (wt. %) samples: (**a**) initial samples; (**b**) L-653 sample; (**c**) L-743 sample; (**d**) H-743 sample.

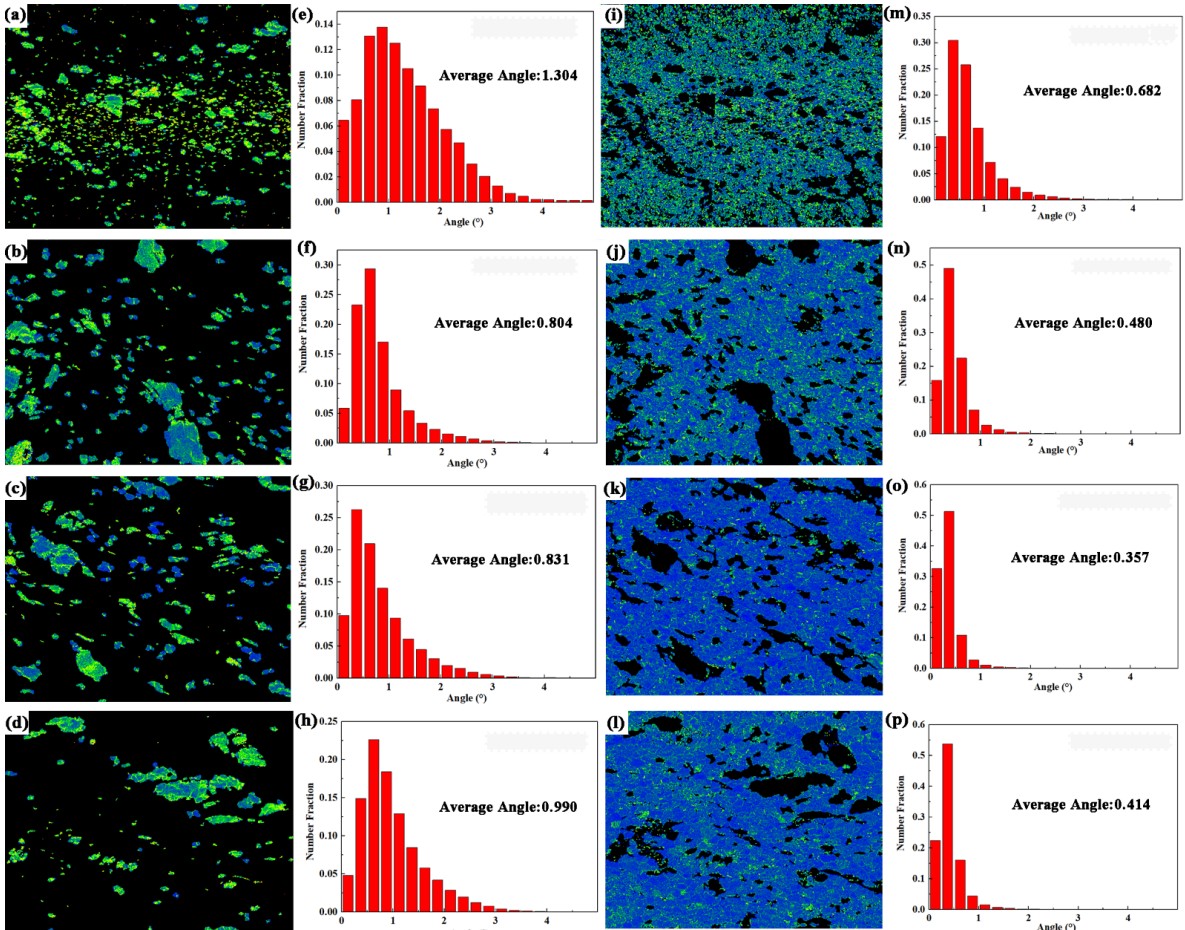

**Figure 15.** KAM maps and the distribution curve of the deformed grains: (**a**,**e**) initial samples; (**b**,**f**) L-653 sample; (**c**,**g**) L-743 sample; (**d**,**h**) H-743 sample. KAM maps and the distribution curve of the DRX grains: (**i**,**m**) initial samples; (**j**,**n**) L-653 sample; (**k**,**o**) L-743 sample; (**l**,**p**) H-743 sample.

3.2.5. Analysis of Re-Dissolution Behavior of the Precipitate Phases

Vasil'ev et al. [38] studied that the lattice of the precipitated phase changed into the matrix lattice (lattice transformation) with the precipitates completely dissolving into the matrix, and another change was the re-dissolved solute atoms entered the matrix by diffusion, which made the supersaturated solid solution more uniform. In the experiment, the precipitated phase -$Mg_5$ (Gd, Y, Zn) (Figure 4) did not change its lattice because $Mg_5$ (Gd, Y, Zn) phase was completely coherent with the $\alpha$-mg matrix [15,17,18]. Thereby, it was re-dissolved by diffusion. The authors of [39] also supported that the solute atoms were uniformly dispersed into the $\alpha$-Mg matrix by diffusion to achieve the re-dissolution of the precipitate phase. The deformation temperature was 653~743 K, and when atoms diffused and dispersed into the $\alpha$-Mg matrix to obtain re-dissolution, this process of diffusion was merely slow. In this experiment, numerous defects (dislocations and vacancies) introduced into the interface between LPSO phase and $\alpha$-Mg matrix, when the completed recrystallized RUE-ed samples continue to deform, provided a thermal diffusion channel for RE atoms to achieve the $Mg_5$ (Gd, Y, Zn) phase re-dissolved. On the other hand, the $Mg_5$ (Gd, Y, Zn) phase was a coarse plate-shaped precipitate in macro scale but a kind of hard and brittle phase, and it was easily broken when the stress was concentrated [40–42]. According to the authors of [43], the vacancy-exchange mechanism of diffusion played a leading role of Mg alloys, diffusion was realized by the transition of atoms to adjacent vacancy lattice positions. The crystal of Mg alloys has a certain equilibrium vacancy concentration at any deformation temperature, and the vacancies moved positions continuously. The existence and movement of these vacancies created conditions for RE atoms diffusion. When the RUE-ed samples deformed, massive vacancies were introduced in the $\alpha$-Mg matrix and the movement of the dislocation created a large number of vacancies which promoted the re-dissolution of the $Mg_5$ (Gd, Y, Zn) phase [41,43]. Especially when the broken $Mg_5$ (Gd, Y, Zn) phase under the high strain rate occurred, the interface between the $Mg_5$ (Gd, Y, Zn) phase and the $\alpha$-Mg matrix increased, accompanied by many vacancies. In summary, the multiply of vacancies greatly increased the diffusion rate of atoms. It also explained why the $Mg_5$ (Gd, Y, Zn) phase re-dissolved faster at high strain rate (Figures 8 and 9). In general, with the increase of temperature and strain rate, more energy was provided for the diffusion of atoms, and the re-dissolution of the $Mg_5$ (Gd, Y, Zn) phase was promoted (Figure 9). Due to the $Mg_5$ (Gd, Y, Zn) phase re-dissolved, the pinning effect on the grain boundary weakened, the grain orientation tended to be consistent, and the DRX grains growth and the texture strengthened(Figures 12 and 13).

**4. Conclusions**

In this paper, the stress-strain response and the microstructure evolution of Mg-13Gd-4Y-2Zn-0.4Zr (wt. %) samples obtained by RUE-ed were studied by hot compression. The re-dissolution of the second phase and the evolution of the texture at different conditions was analyzed. The specific conclusions were as follows:

(1) The average grain size of the initial material was 3.02 μm, include $\alpha$-Mg, $Mg_5$ (Gd, Y, Zn) and LPSO phase.

(2) The true stress-true strain curve showed obvious work hardening and softening stages. And the thermal activation energy of the RUE-ed material was: $Q = 150 \pm 1$ kJ/mol and the constitutive equation was: $\dot{\varepsilon} = 1.80 \times 10^9 [sinh(0.0174\sigma)]^{2.47} exp\left[-\frac{150 \times 10^3}{8.314 \times T}\right]$

(3) Re-dissolution of the Mg5(Gd, Y, Zn) phase was appeared in the RUE-ed samples subjected to hot compression deformation. The movement of the dislocation stimulated the re-dissolution of the Mg5(Gd, Y, Zn) phase. The re-dissolution of Mg5(Gd, Y, Zn) phase promoted texture strengthening and DRX grains growth.

(4) The transformation and kinking of LPSO phase played an important coordinating role in the process of hot compression: 18R-LPSO was changed to 14H- LPSO phase at low strain rate while the LPSO phase kinked dominant to coordinated deformation at high strain rate.

**Author Contributions:** Date curation, Writing—Review, Editing, Writing—Original Draft, Z.Z. (Ziwei Zhang); Conceptualization, Methodology, J.Y., M.M., Y.X. and X.L.; Supervision, Resources, Z.W. and H.H.; Investigation, Z.Z. (Zhiming Zhang). All authors have read and agreed to the published version of the manuscript.

**Funding:** Project supported by the Joint Funds of the National Natural Science Foundation of China, grant number U20A20230, the Natural Science Foundation of Shanxi Province, grant number 201901D111176), the Bureau of Science, Technology and Industry for National Defense of China, grant number WDZC2019JJ006, the Key R&D program of Shanxi Province (International Cooperation), grant number 201903D421036, the National Natural Science Foundation of China grant number 52075501, Shanxi Scholarship Council of China, grant number 2021-127, and Scientific and Technological Innovation Programs of Higher Education Institutions in Shanxi, grant number 2018002.

**Institutional Review Board Statement:** Not applicable.

**Informed Consent Statement:** Not applicable.

**Data Availability Statement:** Data is contained within the article.

**Conflicts of Interest:** The authors declare no conflict of interest.

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
