# Peer review of "The Microstructure Evolution of Mg-RE Alloy Produced by Reciprocating Upsetting Extrusion during Hot Compression"

_metals, doi:10.3390/met12050888_

Round 1

Reviewer 1 Report

Dear authors,

thank you for your work.

I think that your work will have a clear scientific added value after a careful revision.  

I cannot accept your work in its present form, as I think there is still some room for improvement.

Some points:

  1. The abstract promises a study of the effects of the process parameter of reversing upsetting extrusion, but the deformation behavior of the material was investigated. I recommend strongly revising the abstract and title.
  2. For me, the motivation for this work is missing in the abstract as well as in the introduction. Why do I need this activation energy, for example?
  3. Hot should not be bold in the abstract.
  4. What is RUEed, SPD? You must introduce each abbreviation before you use it. Stick to one writing for the abbreviation, now you always use RUE-ed or RUEed.
  5. In the introduction, you write excellent density. However, this is always a question of application, e.g. when it comes to ballistic bullets or weights, magnesium with a very low density is not suitable. Please indicate the case for which the low density may be excellent.
  6. …, average density of magnesium is one thirds lower than that of 27 aluminum (Al) alloys and one fifth of that of steel [1-2]. … Please add approximately in the sentence. …, average density of magnesium is approximately one thirds lower than that of aluminum (Al) alloys and one fifth of that of steel [1-2]. …
  7. The following sentence contains some errors in content: “Rare earth elements can be used as an excellent magnesium alloy modifier to improve the thermal stability of magnesium alloy at room temperature and high temperature, due to the atoms of rare earth elements have a special structure of electrons outside the nucleus and the atomic size is similar to that of magnesium.” 1. “thermal stability of magnesium alloy at room temperature” This is a contradictory statement. 2. The reason for the increase in high temperature strength is the formation of stable phases and not the size of the atoms or the position of the electrons. The position of the electrons in the atom allows them to form similar compounds. However, Gd is rather an unsuitable alloying element due to its high solubility in Mg. 3. Deleted this part of the sentence: “… the nucleus and the atomic size is similar to that of magnesium.“ In principle, almost all alloying elements with a solubility in Mg meet this point, since otherwise there is no solubility in the hpc.
  8. They should tighten up the introduction a bit in relation to their current work.
  9. There is always a space between the units and it is not KV it is kV.
  10. Check all the text because there are some spaces missing.
  11. Also stay with one unit for temperature; you switch between K and °C.
  12. For the repetitive upsetting and extrusion (RUE), I am missing the speed during deformation, like the extrusion speed.   
  13. This sentence is somewhat wrong: “The temperatures were 380℃, 410℃, 440℃, 470℃; strain rates were 0.001s-1, 0.01s-1, 0.1s-1, 0.5s-1, respectively. the engineering strain was 0.8.” Furthermore, it is not clear to me what the technical elongation of 0.8 without a unit is supposed to tell me.
  14. In general, it is not clear to me in the experimental part what and how the process with the parameters will be. How many times was the cycle (Fig 1. (a)) done? What are D and H after the upsetting? When I first read your work, I thought the temperatures and strain rates were the manufacturing parameters. However, they are the test parameters for the compression tests, because my expectation at the beginning was a parameter study and not a study of the material properties. You should make this clearer in the text.
  15. What was the direction of the samples? I have no direction for example in Fig. 2 and for Fig. 3 I have a plane but not the direction in relation to the figure (process to the left or right or up?). I can guess but this should not be the case, you should add it for a good interpretation.
  16. It is not “Φ” or “after corrosion” for diameter, it is “Ø”.
  17. It is not “corrosive agent” or “after corrosion” in line 104 and 105, it is “etchant” and “after etching”.
  18. I miss the unit for the step size and the time for the XRD measurements. Furthermore, I believe that the diagrams (Fig 3. (c) and Fig. 10) from these measurements should be displayed differently. My suggestion is to scale the y-axis logarithmically. At the moment the small peaks are hardly or not at all recognizable, maybe it improves the visibility or the quality of the measurement is not sufficient. Especially in Fig. 10 the background noise seems to be about the same as the peak size. Where do you get the data for the diffraction reflections? Please add the source.
  19. “3. Initial microstructure” belongs in the results section for me. However, your results section is a result and discussion section.
  20. In line 144 it is not Fig.4(c), it is Fig.4 (d).
  21. Can you use the same scale in fig. 5 if the lines in (a) and (b) are not too close to each other. If this is not possible, the scale should be the same for (a) and (b) and also for (c) and (d).
  22. 2.: Use the same number of comma digits everywhere. (Here the temperature is in °C and in Fig. 5 in K.)
  23. I am not sure if the term "rheological softening" is correct and generally understood. If you could introduce the basic effect in the paper, this would make the work accessible to a wider audience.
  24. I think you should introduce indices for the three ε̇ because the values will not be identical.
  25. The indices of A is missing in formula 4, 5 and 6.
  26. Use the same unit for all activation energy Q=287380 J/mol -> Q=287.380 kJ/mol. And it is a small k for kilo (10^3) and not K for temperature.
  27. In Fig. 6 (a) and (b) the labelling of the x-axis is not visible.
  28. The following statement is not true: “By transforming formula (6), the calculation formula of deformation activation energy was shown as formula (7).” You have transformed and derived formula 6. Would you please explain these steps in a more comprehensible way. Perhaps I am not a suitable reviewer on your subject, but it is not clear to me how you arrive to Q (150±1 kJ/mol) and what and where the Q2 in Fig. 6 (c) comes from.
  29. It is the Zener-Hollomon parameter and not Zener-Hollom”a”n parameter.
  30. The conversion of Fig. 9, Fig. 11 and Fig. 12 into PDF format seems to contain errors. Some pixels have been shifted.
  31. In Fig. 10 only the LPSO phases are listed, but the introduction mentions different types of LPSO phases. Can you use your techniques to determine the type of LPSO phases and include them? In the abstract you also state that these are the two LPSO phases, but at the moment it is not clear to the reader how you came to this conclusion. At least not to me.
  32. In Fig. 11 they write of yellow arrows showing the kink, but I think they mean yellow boxes. I do not have yellow arrows in my image.
  33. You have not introduced the abbreviations (a) L-380 sample (b) L-470 sample (c) H-470 in Fig. 12. Could you add to it or better just write the strain rates and temperatures in a generally understandable way. L is probably low speed and H is high speed, but they have several high or low speeds. If I understand it correctly, the texture rises with a higher temperature and falls again with a higher strain rate. Can you explain this?  For me it has to be clarified whether the effect between (b) L-470 sample and (c) H-470 sample or (a) L-380 sample is due to the very different measuring area.
  34. Worng „V ASILS et al“, it is “ S. Vasil’ev et al.” or “Vasil’ev et al.”.
  35. Why are there two conclusions? The first one is a summary, but there are some points in it that were not really discussed in the results and discussion part. In this summary, the 5 of Mg5(Gd;Y; Zn) should be subscript.
  36. In general, I am missing a discussion on how the temperature and strain rate influence the texture and thus also the microstructure. When both increase, there is more energy in the system which leads to increased diffusion and thus promotes recrystallization as well as texture changes. Mg5(Gd;Y; Zn) is, in my opinion, not visible in any of the images, therefore some images with a higher magnification are missing here in which one can clearly see these precipitates.
  37. The references are neither correct nor consistently wrong nor complete nor correct. You must go through the sources carefully and bring them into the desired format! A few examples:
  38. Wx A , Jy A , Bd A , et al. Deformation temperature impact on microstructure and mechanical properties uniformity of GWZ932K alloy under reciprocating upsetting-extrusion. 2022, https://doi.org/10.1016/j.jmrt.2021.12.090 -> Wrong authors names, no issue, no journal and no page numbers.
  39. Xiangsheng Xia1,2,3,4,Kui Zhang1,2,3,Minglong Ma1,2,3,Ting Li5.Constitutive modeling of flow behavior and processing maps of Mg–8.1Gd–4.5Y–0.3Zr alloy.Journal of Magnesium and Alloys.2020,8(3):917-928, https://doi.org/10.1016/j.jma.2020.02.018 -> Names copied with the numbers
  40. Hagihara, Koji1 [email protected], Zixuan1Yamasaki, Michiaki2Kawamura, Yoshihito2Nakano, Takayoshi3.Strengthening mechanisms acting in extruded Mg-based long-period stacking ordered (LPSO)-phase alloys(Article).Acta Materialia.2019:226-239, https://doi.org/10.1016/j.actamat.2018.10.016 -> Names copied with the email addresses and forgetting spaces.

Reviewer 2 Report

This paper studies the hot deformation of Mg-RE alloy at deformation temperatures ranging from 380 °C to 470 °C and with strain rates of 0.001, 0.01, 0.1, 1, and 5s-1. In addition, constitutive models were developed. The microstructure evolution was analysed before and after deformation process. The present paper is interesting, however, to be accepted for publication the following comments need to be addressed

  • Moderate English changes are required in the revised manuscript

Abstract

  • The aim of this work isn’t clear. Please rewrite the abstract including the main aim of the current study and the most significant results.

Material and experimental details

  • Regarding the temperature units, please unify the used unit; for example, in L87 the authors stated, “diameter was homogenized at 793 K for 12 hours” while in L94-95 stated “The temperatures were 380℃, 410℃, 440℃, 470℃”.
  • How many samples for each condition were used for compression test? Please clarify in the manuscript.
  •  

Results

  • It is recommended to add the part 3 (initial microstructure to the results section).
  • Again, the Legend of the figure 5, the unit of temperature was in K while in the experimental part was in C. please revise this issue carefully.
  • The authors stated that “the engineering strain was 0.8” in experimental part, while the results in Figure 5 does not match this.
  • Regarding the misorientation angle in Figure 14, it is recommended to calculate the LAGB and HAGB fraction for each condition.

Conclusion

The manuscript has two conclusions’ sections, please revised. The section 5 is the discussion part.

Reviewer 3 Report

In the present work the stress-strain behaviour and the microstructure evolution of Mg-13Gd-4Y-2Zn-0.4Zr (wt. %) alloy were evaluated. In particular, the study was focused on hot compressed samples after repetitive upsetting and extrusion (RUE) deformation. Specifically, the effect of (i) Grains size, (ii) Second phase evolution and (iii) Texture evolution was evaluated.

The methodological part is well detailed in all its parts. The discussion of the results is well supported by the reported data and the conclusions appear to be consistent with these. Furthermore, the results collected and discussed are widely supported and compared with the existing bibliography.

In my opinion, the work is very interesting; therefore, worthy of being published following the suggested minor revisions:

  • Considering the architecture of the Gleeble (cooled clamps and center of the sample at high temperature), the authors evaluated any thermal gradients in the compression tests conducted in Gleeble?
  • What is the overall duration of the chemical attacks carried out, after the metallographic preparation?
  • Table 2 shows the peak stress under different deformation conditions with the relative standard deviation values. Not being reported in the text, can the authors specify how many replications were carried out for each experimental condition investigated?
  • Can the authors justify the choice of Temperatures and Strain Rates adopted for the experimental activities? With reference to the maximum temperature (470 °C) and the minimum strain rate (0.001 s-1) investigated, the authors recorded any deformation trends attributable to a superplastic behavior of the alloy?
  • Sections 5 and 6 are named in the same way (“Conclusions”); can the authors correct this aspect? The name of section 5 probably needs to be changed.

Round 2

Reviewer 1 Report

Dear authors,

thank you for your work, but most of the comments are only half-heartedly done.

I cannot yet accept your work in its present form, as I think there is still room for improvement and it is necessary.

Critical points from my point of view:

  1. According to the illustration, it is a closed circuit and therefore the number of runs and the point of extraction is not clear. To make it clear for everyone, you should remove the one arrow in Fig. 1 or write it in the text.
  2. Line 109: You added the sentence "The strain rate was 0.002 s-1.". What do you mean by strain rate here? Your specification only works for me in tensile tests, compression tests and for the upsetting process but not for the extrusion. For me it is a movement in per time, so a unit in mm/s or similar.
  3. Since the transformation from one LPSO phase to the other is a diffusion process, the times for heating the pressure samples should be added. Different heating or holding times would lead to roughly different results in the microstructure.
  4. Feedback on her response to 18: It is not about the scaling but about the relationships in the height of the peaks. The logarithmic presentation makes small peaks appear larger and the reader can recognize them better. And yes, I have read about a.u. before and it stands for arbitrary unit and is common knowledge in our technical report, so it does not need to be explained. Your PDF card should have a source, otherwise it is useless!
  5. I still miss the unit for the step size and still miss the measurement time per step as well as the current of the radiation source for the XRD measurements.
  6. Line 142: I have problems with the term "fully dynamic recrystallization" in relation to Fig. 2. I am not able to see a grain and therefore have problems to confirm this statement. Later, in Fig. 3, I see grains with a rotation of crystallographic orientation. This rotation in a grain is an indication of a not completely recrystallized microstructure. At the same time, I would suggest that in Fig. 3 a) you should use a higher resolution image or choose a higher magnification because the small grains cannot be recognized in this PDF. In Fig. 4 a) has the material “(a) EBSD map of deformed grains” (Line 181).
  7. Responds to “24-25. In the process of constructing the constitutive equation, there are simultaneous formulas. It cannot be modified here.”: An index is not a change in the formula, it is an indication that the mathematical equations are different for an identical or similar problem. You yourself have used indices in equations 1, 2 and 3, as in A, A1 and A2. These indices cannot simply disappear when you convert them to equations 4, 5 and 6. In the result under Fig. 6, you have indices again at n1, n2 and Q2. You have to give uniform and explicit indications and three equations with the same result ε̇ are indistinct and not comprehensible for anyone! Here another complicated thing comes into play, n1 is from Zener-Hollomon or n2 from Arrhenius functions or vice versa. Why not just use nArr and nZH?
  8. Responds to “26. KJ/mol is different from K∙J/mol, So it is no problem to use KJ/mol”: Firstly, the capital K is the unit Kelvin, i.e. temperature, but k should probably stand for thousand or 10^3. Second, Q is J/mol, so 1000 J/mol is 1 kJ/mol. It is best to use identical units in a document. If you cannot, if you cannot convert J/mol and kJ/mol, I doubt the correctness of your whole work! 287380 J/mol = 287,380 KJ/mol.
  9. Responds to “31. Generally, the type of LPSO is not marked when magnesium alloy is characterized by XRD. Moreover, in the manuscript, the two types of LPSO phases are distinguished by morphology. The characterization of LPSO phase is also consistent with the research results of our group. I don't think this part conflicts with what I call lamellar and block- shaped LPSO.” If you cannot distinguish between them, LPSO phases (plural) should be included in the legend of Fig. 10.
  10. There is no separation between the results and the discussion as specified in the instructions.

Less relevant points:

  1. Line 20: I miss the space between “.” And “Numerous”
  2. Line 21, Line 295, Line 297 and Line 30 to 31: Change Mg5(Gd, Y, Zn) to Mg5(Gd, Y, Zn)
  3. Line 19: Add an space between 150±1 and KJ/mol.
  4. Line 35: Please change lightest against low density, because 1 kg of each metal is 1 kg, but the density plays the role here.
  5. Line 44 to 45: Add spaces between “were” and ”10H, 14H,” and “18R, 24R [4-6]”
  6. Line 49 to 51: I am not sure, but it should write alloys instead of alloy in the sentence.
  7. Line 51: To remain consistent with the other abbreviations, you should write Severe Plastic Deformation here. Therefore with capital letters in the full term.
  8. There are still a lot of missing spaces! e.g. Line 60 6.9μm, Line 62 320MPa and 258MPa, Line 84 350-500°C, Line 137 15mm and more. Please use the international standard of value and unit with a space in between. The exceptions are the angle in °, which belongs to the value. As with percentages, the character or unit may or may not be attached to the value, but it should be identical throughout the manuscript.
  9. But at the same time to mutch spaces in Linie 66 Mg5" "(Gd, Y, Zn).
  10. Line 84: Where to the ~?
  11. Line 132, 134 and 136: Change KV to kV.
  12. Line 134: Add spaces between “50”and “kV” and “70”and “kV”.
  13. Line 83: It is “ Mosadegh et al. [20]” or “Mosadegh et al. [20]” and NOT “Mosadegh S. et al. [20]“.
  14. Line 91: I miss a point between “temperature” and “However,”.
  15. Line 131: … composition of phases in magnesium alloys. This sentence suggests several alloys, but you are only examining one alloy. You probably mean material, i.e. an alloy that has different properties due to different temperatures and deformations.
  16. Line 138: Even if ED-TD are commonly used, all abbreviations should be written out the first time you use them.
  17. Line 296: It is (0.001 s-1) or (0.001 1/s). Here you can also see the reason for the blank or space.
  18. B e constant. Change in notation for planes, like the change between <> and (). And also the order should be plane then direction in the manuscript.
  19. You have now explained L and H. However, they have not changed the number that stands for the temperature in °C, this should now be there as Kelvin values.
  20. In Fig. 11 b) you talk about yellow arrows indicating the kink, but I think they mean yellow boxes. I still do not have yellow arrows in my picture. (Line 322)
  21. I prefer identical scaling for comparative diagrams as in Fig. 14.
  22. Wrong no. in the discussion and also in the conclusions section.
  23. Line 417: Wrong temperature 380~470 K.
  24. Still: The references are neither correct nor consistently wrong nor complete nor correct. You must go through the sources carefully and bring them into the desired format! Look at the Template (https://www.mdpi.com/files/word-templates/metals-template.dot) at Instructions for Authors (https://www.mdpi.com/journal/metals/instructions).

Reviewer 2 Report

the authors have addressed my comments. only one note here, the results section, should have the number 3 not 4 in ranking. 

Author Response

Related chapter order problems have been revised in the manuscript.